# Acetic Acid Immersion Alleviates the Softening of Cooked *Sagittaria sagittifolia* L. Slices by Affecting Cell Wall Polysaccharides

**DOI:** 10.3390/foods12030506

**Published:** 2023-01-22

**Authors:** Yangyang Sun, Yanzhao Liu, Jie Li, Shoulei Yan

**Affiliations:** 1College of Food Science and Technology, Huazhong Agriculture University, Wuhan 430070, China; 2Aquatic Vegetable Preservation and Processing Technology Engineering Center of Hubei Province, Wuhan 430070, China

**Keywords:** *Sagittaria sagittifolia* L. slices, acetic acid immersion, cooked texture, immunolabeling analysis, cell wall polysaccharide

## Abstract

This study investigated the mechanism for acetic acid pretreatment to improve cell wall integrity and thereby enhance the hardness of cooked *Sagittaria sagittifolia* L. slices by affecting polysaccharides in the cell wall. Distilled water immersion and 0.6% acetic acid immersion (the solid/liquid ratio is 1:10) for 15 h at room temperature could result in the conversion of pectin through different reactions during thermal processing. Combined in situ and in vitro analysis demonstrated that acetic acid pretreatment could promote the interaction of cellulose microfiber or hemicellulose with RG-Ⅰ side chains during thermal processing of *S. sagittifolia* L. slices, promote the entanglement between linear pectin molecules and make hemicellulose show a lower molecular weight under cooking, making it easy to firmly bind to pectin, which resulted in texture changes. The findings may help improve the texture of thermally processed vegetables and fruits and deep processing of starchy vegetables.

## 1. Introduction

*Sagittaria sagittifolia* L. (arrowhead) is a perennial herb belonging to the Alismataceae family. It is rich in nutrients, such as polysaccharide, protein, fat, vitamins and some mineral substances. Moreover, arrowhead bulbs also show medical value because of their antitumor, antioxidant and antibacterial activity [1]. Arrowhead bulb is suitable for braising, frying and cooking, and this thermal processing can remove unpleasant bitter tastes [2]. However, thermal processing results in softening and thickened soup, and the final texture struggles to meet the demand of the majority of consumers. Thus, it is critical to explore the changes in texture resulting from cooking. Moreover, the regulation of cooked texture is of great significance to the improvement of the edibility and commercial value of arrowhead.

Texture is an important factor influencing sensory attributes of fruits and vegetables. It undergoes quick changes during thermal processing owing to cell wall disassembly [3,4,5]. Acetic acid treatment, as a safe, cheap, environmentally friendly method, can be used to prevent disassembly of the fruit and vegetable cell wall during cooking [6,7,8]. The cell wall is a supramolecular assembly of pectin, hemicellulose and cellulose, among which pectin is the most water soluble during thermal processing. Pectin is constituted by homogalacturonan (HG), rhamnogalacturonan I (RG-I) and rhamnogalacturonan II (RG-II) [9] and is often found in the primary and middle layers of the cell wall, which greatly affects the cell wall adhesion, strength and porosity [10]. Pectin modification including thermal dissolution, β-elimination and depolymerization may occur during cooking and result in the weakening of cell–cell adhesion in the middle lamella, which causes the loss of texture to varying degrees [11,12,13]. However, acetic acid can effectively alleviate the decrease in hardness after cooking by reducing CSF thermal degradation and slowing down the β-elimination reaction [7]. Moreover, Zhao et al. reported that acetic acid immersion pretreatment obtained the highest firmness and most intact structure of the cell wall for cooked potato slices, which may be achieved by increasing the relative contents of CSF and NSF and promoting the pectin–cellulose interaction [6]. Currently, the effect of acetic acid treatment on texture is mainly studied in potatoes and lotus rhizomes. However, the mechanism of the acetic acid affecting the texture of arrowhead has not been revealed, the changes in and interactions of cell wall polysaccharides caused by cooking are not fully understood yet.

Currently, how thermal processing affects the polysaccharide structure of the cell wall in food matrices has been investigated in vitro, but the possible destruction of pectin structure by the extraction process has not been considered, which can easily affect the accuracy of the results. Some monoclonal antibodies (mAbs), such as LM 20, LM 26, LM 6-M, can be used as a very powerful tool to provide information about defined pectin domains in the cell wall and the potential interactions of polysaccharides [14,15,16]. Based on previous reports, mAbs were used to observe broccoli, tomato, carrot and strawberry pectin [17,18,19]. This may complement the lack of in vitro characterization methods of pectin. To date, in situ research on pectin with mAbs has been not conducted in arrowhead bulb slice processing.

In this research, the influence of acetic acid immersion on the texture, microstructure and cell wall polysaccharides of arrowhead bulb slices during cooking was analyzed and compared by combining immunolabeling and physicochemical analysis of isolated cell wall material from arrowhead bulb slices, so as to identify the variations of cell wall polysaccharides and their relationship with the cooked texture of arrowhead bulb slices. Our results may provide a theoretical foundation for improving thermally processed starchy vegetables and fruits in terms of quality.

## 2. Materials and Methods

### 2.1. Materials and Sample Preparation

Arrowhead bulbs were obtained from a local supplier in November 2021 in Guangxi, China. Following washing with running water, the bulbs were peeled, and then made into 5 mm slices for subsequent experiments. The slices were randomly divided into two groups: (1) 200 g slices pretreated immersion in 2000 mL of distilled water for 15 h (DWI), and (2) 200 g slices pretreated immersion in 2000 mL of 0.6% acetic acid (*v*/*v*) for 15 h (AAI). Then, 100 g slices from the DWI or AAI group with 130 mL distilled water were vacuum-packed in Nylon pouches (24 cm × 17 cm) and placed in boiling water for 0.5 h, followed by rapid cooling to room temperature with running water, which were named as cooked DWI (DWIC) and cooked AAI (AAIC), respectively. Some of the samples were used for texture measurement, SEM observation and immunolabeling analysis, and the remaining slices were kept at −80 °C for further experiments. The monosaccharide standards, namely galacturonic acid (GalA), rhamnose (Rha), galactose (Gal), mannose (Man), arabinose (Ara), glucose (Glc), fucose (Fuc) and xylose (Xyl), were purchased from Yuanye Bio-Technology Co., Ltd. (Beijing, China). All chemicals used were of analytical grade and purchased from reliable reagent dealers.

### 2.2. Texture Measurement

The hardness of samples was measured with a TA.XTplus texture analyzer (England Instrumentation System Co., TA London, UK) equipped with a P/6 probe, following the descriptions of Zhao et al. [6] with some modifications. The samples: 5 mm arrowhead bulb slices with different treatments. The measurement mode was TPA. The pretest speed, testing speed and posttest speed were set at 3.00, 0.50 and 3.00 mm/s, respectively. The compression ratio was 30%. Time between two compressions was 2 s. The maximum peak value at the first compression was used to represent the hardness. Analysis of each sample was conducted in nine biological replicates.

### 2.3. Scanning Electron Microscopy (SEM)

SEM (HITACHI SU-8010, Tokyo, Japan) was employed to visualize the shape and surface morphology of the cell wall of arrowhead bulb slices under different treatment conditions. Arrowhead bulb slices were further cut into smaller pieces and fixed in 2.5% glutaraldehyde. Then, the pieces were treated with ethanol in a sequence of concentrations (30%, 50%, 60%, 70%, 80%, 90% and 100%) after washing three times (5 min for each washing) with 0.1 mol/L phosphate buffer (pH 7.2). The samples were then subjected to critical point-drying with liquid CO_2_ and gold-coated with gold sputter. Finally, sample observation was performed with SEM at an electron acceleration voltage of 5 kV under high vacuum [7].

### 2.4. Immunolabeling of Pectic Epitopes

The mAbs can complement the deficiency of the in vitro pectin characterization method through in situ visualization of pectin by antipectin mAbs [17]. Pectin was labeled in situ with antipectin mAbs, and variations of fluorescence intensity of the cell wall were examined with confocal laser scanning microscopy (CLSM) to accurately analyze how acetic acid pretreatment affects the cell wall pectin domain. The primary antibodies, including methyl-esterified homogalacturonan (LM 20), antibranched 1, 4-galactan (LM 26) and LM 6-M (with greater avidity of binding to Ara) [16,17,20], were obtained from Kerafast Inc., Boston, MA, USA. The secondary antibody was antirat Ig and FITC purchased from Sigma-Aldrich, Shanghai, China. In situ immunolabeling of pectin epitopes was carried out according to the procedure reported in a recent study [21]. Arrowhead bulb pieces (5 mm × 5 mm × 20 mm) were fixed with 3% agar solution and then cut into 100 μm slices with an oscillating tissue slicer (Leica vt1000s, between Cologne and Frankfurt, Germany). The primary antibodies were diluted by 20-fold with 3% skimmed milk powder, and then the slices were transferred to diluent for 1 h at room temperature. Subsequently, the slices were washed with phosphate buffer solution (PBS, Ph 7.2) three times (5 min for each washing). The slices were transferred to the antirat Ig and FITC (the antibody was diluted by 40-fold with 3% skimmed milk powder) to carry out the secondary labeling and subjected to inoculation for 1 h at room temperature, and washing was carried out as described above. The slices were observed with an Olympus FV3000 microscope. Different samples were immunolabeled at least in duplicate.

### 2.5. Separation of Cell Wall Fractions

The collected materials were freeze-dried and processed into powder using a blender (MX-101SG1, Panasonic Corporation, Minzen city, Osaka, Japan), and then starch was removed with alpha-amylase and amyloglucosidase [15]. About 100 mL of ethanol (95%, *v*/*v*) was supplemented to the residue, which was stirred for 20 min in 85 °C water to remove soluble sugars, and the mixed solution at a methanol–chloroform volume ratio of 1: 1 was used to remove the lipid. Finally, alcohol-insoluble residue (AIR) was obtained.

Following the procedure of Zhao et al. [6], cell wall polysaccharides were sequentially extracted from AIR to obtain the water-soluble fraction (WSF), chelate-soluble fraction (CSF), sodium carbonate-soluble fraction (NSF) and hemicellulose fraction (HF). Subsequently, the obtained solution was subjected to at least 48 h of dialysis in distilled water using a 3500 Da dialysis bag followed by evaporation (RE-52AA rotary evaporator, Shanghai Yarong, Shanghai, China). The concentrated solution was precipitated using 95% ethanol in five-fold volume and then freeze-dried for further experiments.

### 2.6. Fourier Transform Infrared (FT-IR) Spectroscopy of the AIR

An FT-IR spectrometer (Nicolet iS 50, Thermo Fisher Scientific Inc., Massachusetts, USA) with KBr discs was used to collect the infrared spectra of AIR. According to Zhang et al. [22], samples were prepared by mixing AIR powder with KBr at the rate of 1:50 (*w*/*w*) and pressing into pellets. Then, they were scanned over 4000–500 cm^−1^ at a 4 cm^−1^ resolution 64 times against KBr as the background. The data were analyzed with the Omnic 8.2 software (Thermo Fisher Scientific, Inc., Madison, WI, USA).

### 2.7. Monosaccharide Composition Measurement in Fractions

The monosaccharide composition was measured with a previously described method by Liu et al. [7]. Cell wall fractions from different treatment groups (10 mg), 2 mL of distilled water and trifluoroacetic acid (4 mol/L) were added to an ampoule, and then hydrolyzed at 110 °C for 4 h. The samples were nitrogen-dried and then diluted to 1 mL with ultrapure water. The monosaccharide in samples was derivatized with 1-phenyl-3-methyl-5-pyrazolone (PMP). Finally, the monosaccharide solution labeled with PMP was passed through a 0.45 μm membrane and analyzed with high-performance liquid chromatography (HPLC, Waters e2695, Massachusetts, USA.).

The HPLC analysis conditions included a C18 column (Agilent ZORBAX Eclipse Plus C18). This system used 0.1 mol/L sodium phosphate buffer solution (pH 7.2) and acetonitrile as mobile phases, and the a:b ratio was 83:17 (*v*/*v*). The flow rate was 1 mL/min; the injection volume was 10 μL; and the detection wavelength was 245 nm.

The sugar ratio was calculated based on a previously described method [23]. Sugar ratio 1 represented the linearity of pectin, which was expressed as the galacturonic acid (GalA) ratio in the sum of neutral sugars in pectin side chains. Sugar ratio 2 was obtained by calculating the ratio of Rha/GalA, which could indicate the contribution of RG to the pectin population. Sugar ratio 3 was calculated as the ratio of (Ara+Gal)/Rha, and used to represent the branching extent of the RG-I.

### 2.8. Molecular Mass Distribution of Cell Wall Fractions

By using the previously described method by Zhang et al. [22], the polysaccharide molecular mass distribution in different fractions was analyzed through gel permeation chromatography (GPC, 1100 LC/MSD Trap, USA Instrumentation System Co., Agilent, California, USA). Two columns (shodex OHpak SB-804 and shodex OHpak SB-806) connected in series were employed to elute polysaccharides at 0.3 mL/min and 25 °C which were then detected by a refractive index detector. Elution was performed with 0.1 mol/L NaNO_3_ (filtered through a 0.45 m filter). Pectin solutions (0.3 mg/mL) dissolved with eluent were filtrated through 0.45 μm membranes before use. A dn/dc value of 0.147 mL/g was employed for the molar mass analysis of cell wall fractions. Standard dextran with 40 kDa molecular mass was used for normalization.

### 2.9. Thermogravimetry Analysis

Thermogravimetry (TG 209 C, Netzsch, Selb, Germany) was used for thermal analysis according to Zhang and Wang [24] with some modifications. Samples (5 ± 0.1 mg) were accurately weighed in standard aluminum pans, which were then sealed and analyzed using TG. The pans were scanned from 40 °C to 500 °C (heating rate = 10 °C/min) with an empty aluminum pan as the reference. All runs were carried out at least twice.

### 2.10. Statistical Analysis

All data were presented as mean ± standard deviation, and analyzed with SPSS Statistics 25 for Windows (SPSS, Inc., Chicago, IL, USA). One-way analysis of variance (ANOVA) with Duncan’s test was performed to compare the means. Differences were considered to be significant at *p* < 0.05. All figures were constructed by Origin 2017 (OriginLab, Northampton, Massachusetts, USA).

## 3. Results and Discussion

### 3.1. Effects of Acetic Acid Pretreatment on the Hardness of Fresh and Cooked Arrowhead Bulb Slices

The texture of plant-based foods plays an important role in their processing properties, in which hardness is one of the most important features considered by the consumers [25]. Figure 1 shows that DWI had the highest (4452 g) hardness among samples from the four different treatment groups. AAI had a lower hardness (3136 g) than DWI, possibly due to a drop in osmotic pressure owing to water loss in the cell caused by acetic acid immersion. The hardness significantly decreased after boiling for 30 min (DWIC and AAIC). DWIC showed significantly lower hardness (522 g) than AAIC (1243 g). These results indicated that acetic acid pretreatment could effectively slow down the decline of tissue hardness during cooking, and improve the thermal stability of arrowhead bulb slices. Zhao et al. also reported that acetic acid pretreatment can improve the hardness of potatoes [6]. These findings could be attributed to the decrease in intercellular adhesion and variations in the composition and structure of cell wall polymers [13,14].

### 3.2. Effects of Different Treatments on Cell Wall Microstructure of Arrowhead Bulb Slices

The thermally processed texture of vegetables and fruits is affected by cell wall integrity, cellular turgor and other factors, and cellular turgor disappears quickly after thermal processing [26]. Therefore, the structural integrity, size and arrangement of cells, particularly the integrity and cell wall thickness and separation of adjacent cells, directly determine the tissue texture characteristics [8,12]. We observed the microstructure of arrowhead bulb slices under different treatment conditions by SEM to investigate the association between texture and cell wall. As shown in Figure 2, the fresh samples (AAI and DWI) showed rupture of cells and appearance of starch granules; in contrast, cavities and tiny holes in the cell wall and gelatinization of starch were observed in DWIC, which is consistent with the results of Bordoloi et al. [27]. For AAIC, the starch granules were completely gelatinized and aggregated to fill the interior of the cell, and there was an obvious separation between starch and cell wall. The cell wall structure of DWI and AAI was still maintained relatively intact, particularly that of AAI. Nevertheless, the cell wall structure was slightly damaged in AAIC, while completely destroyed in DWIC. It can be concluded that cell wall integrity is closely associated with the tissue hardness, and acetic acid pretreatment can effectively reduce the cell wall destruction during cooking, thus increasing the hardness of the slice. The same phenomenon has been found in other studies [6,8].

### 3.3. In Situ Immunolabeling of Pectic Epitopes

Figure 3 shows the CLSM images of slices from different treatment groups with different antipectin mAbs.

LM 20, with strong binding ability to methylesterified pectin, was used to label the slice samples. The polygonal cell wall outline was better preserved in fresh slice samples, but disappeared in cooked samples. LM 20 epitopes were identified in cell junctions, and were present at the corners of intercellular spaces in high abundance in fresh samples. However, LM 20 epitopes were only identified in cell junctions in cooked samples, which is similar to the results reported by Christiaens et al. [17]. Generally, a weak cell adhesion will result in cell separation [16]. In this study, CLSM observation of cooked samples showed the occurrence of cell separation. Moreover, AAIC had higher fluorescence intensity than DWIC; in DWIC, pectin polymers recognized by LM 20 appeared to be loosely attached to the cell wall, and dispersed or no LM 20 epitopes were observed at the initially clear spots in cell junctions and the corners of intercellular spaces. These results may be due to the fact that thermally induced β-elimination leads to the dissolution of pectin from the cell wall [28], resulting in the labeling of less pectin by LM 20, but acetic acid pretreatment can effectively reduce the dissolution of pectin during heating.

Redgwell et al. [28] demonstrated that the loss of pectic galactans and arabinan depolymerization is a general feature of texture softening. The arrowhead bulb slice cells in Figure 3 were labeled with LM 6-M and LM 26 antibodies, which is specific to RG-I side chains. As LM 6-M showed a better binding ability to arabinan than LM 6 and preferentially binds to six 1, 5-arabinosyl residues [16], it was used to characterize the arabinan side chains of RG-I. LM 6-M epitopes were distributed in cell junctions, and fluorescence was detected in the cell interior of cooked samples, indicating the dissolution of some arabinan side chains. The fluorescence intensity in AAIC was lower than that in DWIC, which agrees with the high content of arabinose in HF (Table 1) strongly bound to the cell wall after KOH extraction [28]. A possible explanation for this result could be the interaction of the arabinan side chain with the cellulose microfiber during thermal processing after acetic acid pretreatment, and not all arabinose sites were detected in these samples [29].

LM 26 labeled branched 1, 4-galactan (one part of RG-I) and was observed in the cell wall with weak fluorescence. DWI had a higher labeling intensity than AAI, which was in agreement with the monosaccharide composition (Table 1). In addition, the LM 26 branched galactan epitope was relatively increased in the cell wall of cooked samples, particularly in DWIC. It should be noted that after cooking, the content of Gal was higher in AAIC (53.06) than in DWIC (39.97) (Table 1), but the fluorescence of LM 26 in AAIC was lower than that in DWIC. The structural analysis of ripe strawberry cell wall revealed that the side chain Gal interacts with cellulose microfibers to some extent [20]. Therefore, redistribution of Gal in the cell wall may occur during the cooking process [30]. Consistently, Imaizumi et al. [31] observed pectin self-assembly network structure by AFM in bleached carrots.

### 3.4. FT-IR

FT-IR spectroscopy can effectively demonstrate the atom or functional group vibration in polysaccharides [32]. It was used to further characterize the structure of AIR under different treatments. The peaks at 1734 and 1650 cm^−1^ are attributed to the absorption of esterified carboxyl (-COOR) and ionized carboxyl groups (-COO^−^), respectively. The ratio of the peak area at 1734 cm^−1^ to the sum at 1734 and 1650 cm^−1^ could be utilized to quantify the methyl esterification degree (DM) of pectin [25]. The peak at 1734 cm^−1^ disappears in cooked samples, which is contradictory to the CLSM. This result may be due to the ratio of esterified pectin and AIR. The peak intensity at 1650 cm^−1^ was increased in cooked samples, indicating the occurrence of demethylation during cooking [25]. The broad absorption peak at around 3380 cm^−1^ is assigned to the stretching vibration of hydroxyl groups, and the weak absorption peak at 2925 cm^−1^ could be ascribed to the C-H resonance of methyl groups in the sugar ring.

Differences between AIR were also observed in the fingerprint region of 1200–950 cm^−1^. The peak at 1110 cm^−1^ was observed as a typical absorption band of polygalacturonic acid. The AAI and DWI FT-IR spectrum exhibits a stronger peak with a maximum at around 1110 cm^−1^, but AAIC and DWIC show lower absorption at 1110 cm^−1^, indicating degradation of polygalacturonic acid during cooking. The peaks at 1162, 1050 and 1434 cm^−1^ were observed in all samples, which could be assigned to the skeletal vibration of a C-O-C pyranose ring and -CH_2_ scissoring of cellulose, respectively [33]. There was no significant change in the intensity of these peaks, indicating that the chemical structure of the cellulose skeleton remains stable after heating treatment.

### 3.5. Composition and Sugar Ratio of Different Cell Wall Fractions

The sugar composition and sugar ratio were analyzed for polysaccharides to compare the main chain HG and RG-Ⅰ side chain to further analyze polysaccharide structure changes in the cell wall under different treatments. Table 1 shows the sugar composition and sugar ratio of different fractions including WSF, CSF, NSF and HF, which included pectin-related rhamnose (Rha), galactose (Gal) and arabinose (Ara), fucose (Fuc) and galacturonic acid (GalA) and non-pectin-related glucose (Glc), xylose (Xyl) and mannose (Man). Similar results were obtained in another study [32], suggesting that the fractions obtained from arrowhead bulb slices are heteropolysaccharides with different chemical compositions. The Glc, which should not be found in WSF, CSF, and NSF, might be derived from starch or soluble glucomannan not completely removed before sequential extraction [7]. Xyl is a dominant non-pectic saccharide in the cell wall. It was detected in almost all cell wall fractions, particularly in the HF fraction, which was probably derived from soluble arabinoxylan or xyloglucan [16]. The Man found in WSF, CSF and NSF may be generated from hemicellulose or cellulose that is bound to pectin side chains [34]. The components co-eluted with pectin might be tightly connected with cellulose microfibrils through hydrogen bonds. The content of galactan and araban in AAIC was higher than that in DWIC, which might be associated with cell wall firmness and elasticity, respectively [35].

WSF was loosely attached to the cell wall material [36]. CSF is released from the middle lamella, and contains calcium-bound pectin. NSF is released from the primary cell wall [6], and possibly contains pectin that is covalently bound to other polysaccharides in the cell wall. As shown in Table 1, in cooked samples, DWIC had higher GalA contents in WSF but lower GalA contents in CSF and NSF compared with AAIC. This may be due to the occurrence of β-eliminating depolymerization during cooking, resulting in the conversion of CSF and NSF to WSF, which is easier to extract [37]. The pH of distilled water and acetic acid solution is around 6.40 and 3.50, respectively. A high pH improves the occurrence of β-elimination reactions and the depolymerization of pectin is the least dramatic at pH 3.5 [5].

In these fractions, the Ara and Gal contents of AAIC (Ara: 73.03; Gal: 53.06) were significantly increased relative to those of DWIC (Ara: 39.86; Gal: 39.97), indicating that acetic acid pretreatment could result in the degradation of RG-Ⅰ side chains by acid hydrolysis during cooking and the transformation of CSF and NSF into soluble pectin (WSF) [38,39]. The Ara and Gal contents in NSF were significantly higher in AAIC than in DWIC, which is due to more side chains in AAIC than in DWIC (sugar ratio 3). Multi-branched pectin is likely to form complexed cross-linked structures [40]. They are combined with hemicellulose or cellulose more closely and not easily degraded, which is in line with the results of antibody labeling (Figure 4).

The CSF in DWIC had more side chains and fewer main chains, which could hardly contribute to the formation of calcium bridges as a spatial barrier and thus decrease the sample hardness after cooking. In fresh and cooked samples, the content of Glc under acetic acid treatment was significantly increased relative to that under distilled water treatment, implying that acetic acid immersion promotes the interaction between CSF and cellulose or hemicellulose. A similar phenomenon has been found in other studies [7].

Interestingly, HF was composed of Glc and Xyl. The characteristic sugars of pectin, including GalA, Ara and Gal, in HF suggested the co-extraction of the “hairy region”, especially RG-I, with hemicellulose fractions [7]. Therefore, some pectin fractions stay within the tissue and are stabilized by strong bonds [31], and can only be solubilized by highly alkaline solutions. The HF of AAIC contained more abundant arabinose than DWIC. Hence, the RG-I might be mainly linked to hemicellulose through arabinose [37]. It has been indicated that a strong attachment of RG-I to the cell walls may contribute to the mechanical properties of firmer fruits [16].

### 3.6. Molecular Weight Distribution

Pectin matrix is critical in maintaining the cell wall internal structure. Therefore, variations in pectin polysaccharides after different treatments were determined by gel permeation chromatography. Molecular weight (Mw) distribution obtained for fractions from different treatment samples is presented in Figure 5. In general, as can be seen from the Mw profiles, with increasing elution time, the Mw of the eluted components decreased. For WSF, four major peaks were observed in AAI and DWI, but after thermal processing, there were pronounced changes in polymer concentration and Mw distribution pattern in AAIC and DWIC. The peak of region Ⅱ could be separated into two peaks and the peak shifted to the right after cooking, suggesting that heat treatment results in the depolymerization of WSF and breakage of the pectin chain into smaller molecules. The peak of region Ⅰ shifted to the right in DWIC and the intensity of peaks related to high Mw decreased. These results suggested that the WSF extracted from DWIC underwent more serious degradation than that extracted from AAIC, which is opposite to the finding for sugar ratio 1 (Table 1). The WSF of AAIC showed a lower linearity than that of DWIC, indicating the occurrence of entanglement between the linear pectin molecules in AAIC [10]. The same phenomenon was found in HF. In AAI and DWI, one peak was observed in region I for each group, while the peak was separated into two peaks for AAIC and DWIC. The AAIC group had a higher sugar ratio 3, indicating that this group had more branching than the DWIC group. In the HF of AAIC, the peak intensity corresponding to low Mw increased, indicating that the hemicellulose in AAIC was degraded. Then, due to the low Mw, it was easier for hemicellulose to firmly bind to pectin [6], which might be related to the different composition of HF in AAIC. The low Mw molecules are consistent with the Ara and Gal contents in HF (Table 1), indicating the possible presence of branched pectin around hemicellulose. The Mw distribution of the CSF and NSF was similar between AAIC and DWIC, and the Mw of NSF was significantly decreased after cooking.

### 3.7. DTG Analysis

Thermal analysis is important for evaluating thermal parameters of polymers. In the present study, derivative thermogravimetry (DTG) signals were used to explore the thermal degradation of different fractions. The TG curve represents the weight loss (Δm) during heating, and the DTG curve is represented by the first derivation of the TG curve and is associated with the degradation velocity [41]. As shown in Figure 6, the AIRs in different samples all had a wide peak width, indicating that AIR is a multi-component substance and its degradation is a continuous process. The curve of AIR is characterized by the presence of a large exothermic peak with the maximum exothermic velocity at around 330 °C, which may be attributed to cellulose [42]. The maximum degradation velocity (v_max_) decreased after cooking. The AIR in DWIC had no obvious characteristic pectin peak, but that in AAIC had a characteristic pectin peak at 270 °C, indicating that the structure was seriously destroyed in DWIC, and the thermal stability of AIR was improved in AAIC.

For WSF and NSF, the first degradation peak between 70 °C and 150 °C can be ascribed to water evaporation. In addition, the DTG curve showed two exothermic degradation peaks at around 250 °C and 330 °C, which could be associated with the degradation of pectin and the cellulose network, respectively [41]. The same phenomenon has been observed in another study [24]. The degradation peak of cellulose appeared after cooking, indicating that interaction of pectin and cellulose may occur during the cooking process. These data are consistent with the immunological results. For CSF, the pectin degradation peak at about 220 °C occurred earlier than that of WSF and NSF, which may be due to the differences in molecular parameters, modification degree and physical state [41].

## 4. Conclusions

This research demonstrates that pretreatment with 0.6% acetic acid solution can significantly alleviate the softening of arrowhead bulb slices after cooking and maintain the structural integrity of the cell wall compared with distilled water pretreatment. The softening of arrowhead bulb slices is mainly due to the solubilization and conversion of pectin. Acetic acid pretreatment promotes the entanglement between the linear pectin molecules and interaction between RG-Ⅰ side chains and cellulose microfibers. This study provides a new approach for alleviating the softening of plant tissues under thermal processing, and reveals the precise structure of cell wall pectin in arrowhead bulb. In future studies, the relationship between the branched chain of RG-Ⅰ and cell wall needs to be further explored.

## Figures and Tables

**Figure 1 foods-12-00506-f001:**
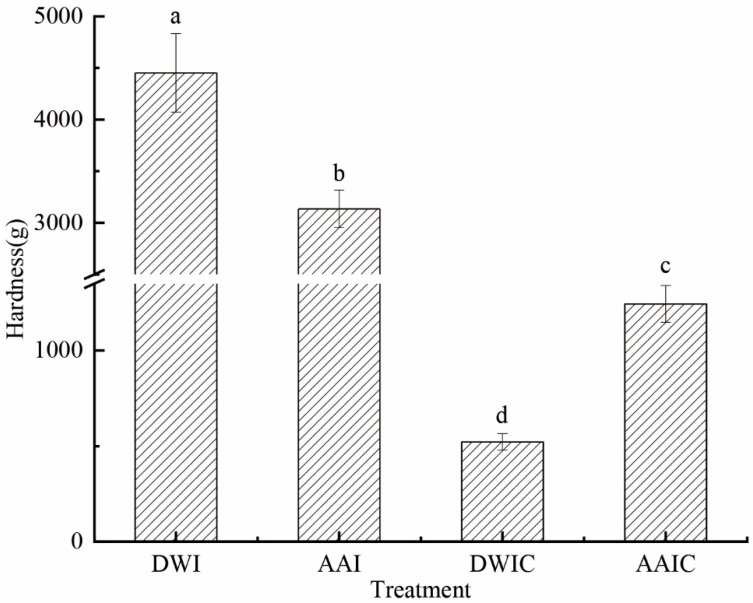
Hardness of arrowhead bulb slices under different treatments. DWI: distilled water immersion for 15 h; AAI: 0.6% acetic acid immersion for 15 h; DWIC: cooked DWI; AAIC: cooked AAI. Data were obtained from nine replicates. Different letters indicate significant differences (*p* < 0.05). The error bar stands for standard deviation (*n* = 9).

**Figure 2 foods-12-00506-f002:**
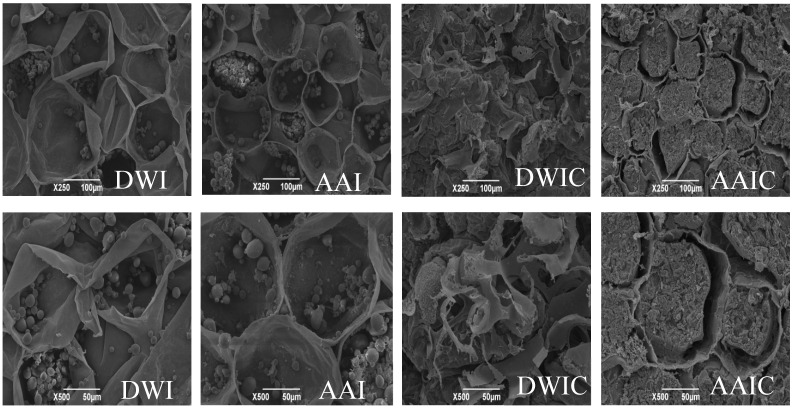
SEM images of arrowhead bulb slices under different treatments. DWI: distilled water immersion for 15 h; AAI: 0.6% acetic acid immersion for 15 h; DWIC: cooked DWI; AAIC: cooked AAI. Each treated sample was observed at 250× and 500× magnification.

**Figure 3 foods-12-00506-f003:**
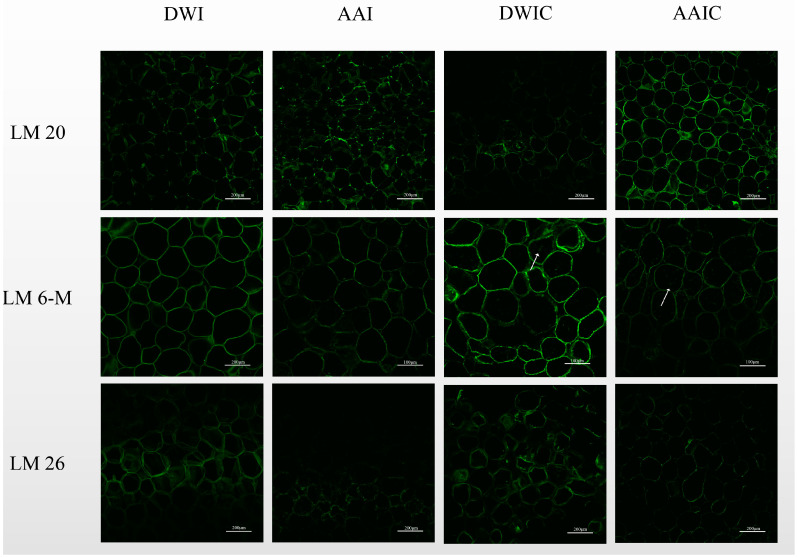
Representative pictures of cell wall immunofluorescence labeling in arrowhead bulb slices under different treatments with LM 20, LM 6-M and LM 26. Arrows indicate the presence of fluorescent sites in the cell interior of cooked samples. The scale bars of LM 20, LM 6-M and LM 26 are 200 μm, 100 μm and 200 μm, respectively. Sections of 100 μm were used. DWI: distilled water immersion for 15 h; AAI: 0.6% acetic acid immersion for 15 h; DWIC: cooked DWI; AAIC: cooked AAI.

**Figure 4 foods-12-00506-f004:**
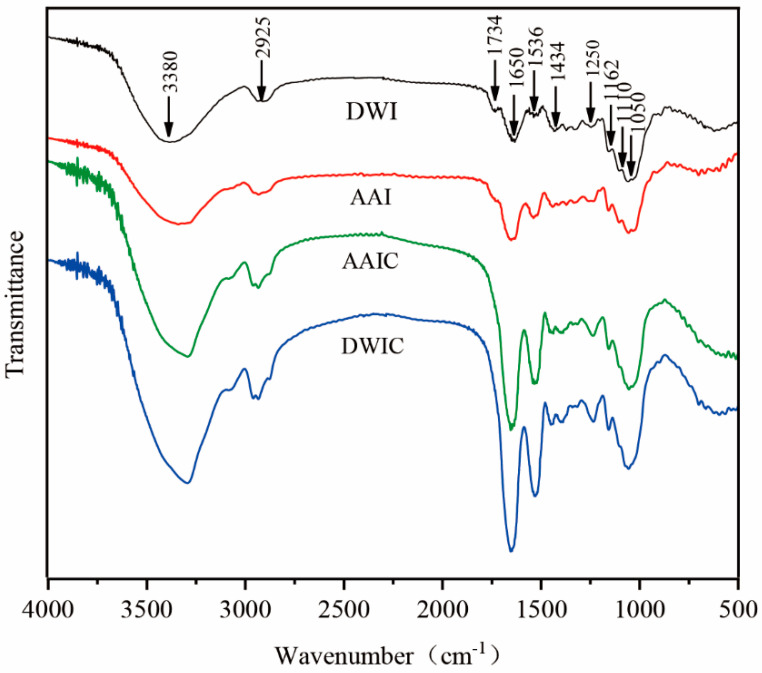
FT−IR spectra of alcohol-insoluble residue (AIR) from different treatment groups in the region of 4000–500 cm^−1^. DWI: distilled water immersion for 15 h; AAI: 0.6% acetic acid immersion for 15 h; DWIC: cooked DWI; AAIC: cooked AAI.

**Figure 5 foods-12-00506-f005:**
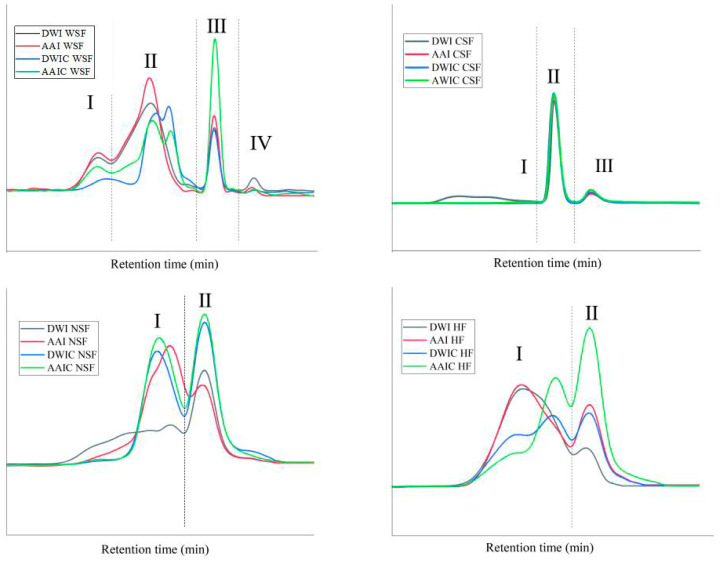
Variations in molecular weight distribution of cell wall fractions between different treatment groups. According to the number of peaks in DWI, we divided the peaks into different regions, which are represented by Ⅰ, Ⅱ, Ⅲ and Ⅳ, respectively. DWI: distilled water immersion for 15 h; AAI: 0.6% acetic acid immersion for 15 h; DWIC: cooked DWI; AAIC: cooked AAI; WSF: water-soluble fraction; CSF: chelate-soluble fraction; NSF: sodium carbonate-soluble fraction; HF: hemicellulose fraction.

**Figure 6 foods-12-00506-f006:**
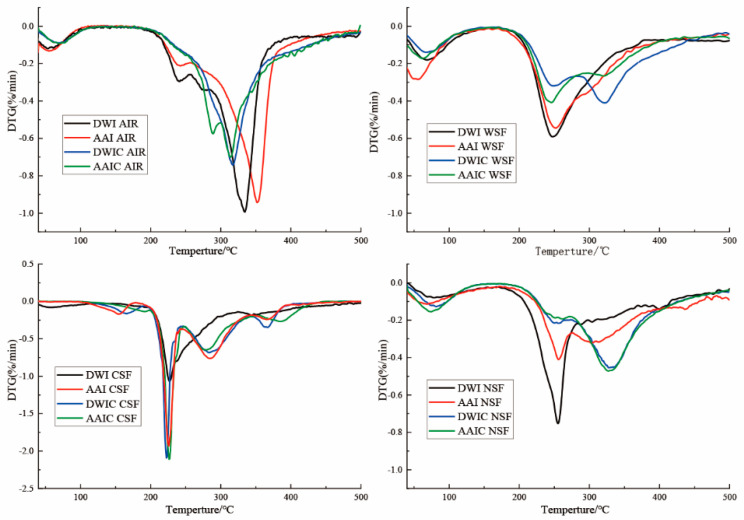
DTG analysis of cell wall fractions and AIR in different treatment groups. DWI: distilled water immersion for 15 h; AAI: 0.6% acetic acid immersion for 15 h; DWIC: cooked DWI; AAIC: cooked AAI; WSF: water−soluble fraction; CSF: chelate−soluble fraction; NSF: sodium carbonate−soluble fraction; HF: hemicellulose fraction.

**Table 1 foods-12-00506-t001:** Monosaccharide composition and sugar ratio (mol%) of cell wall fractions in different groups. Different letters of the same fraction represent significant differences in the same sub-column (*p* < 0.05). Sugar ratios allow the gaining of information with reference to polysaccharide structure from sugar composition data. DWI: distilled water immersion for 15 h; AAI: 0.6% acetic acid immersion for 15 h; DWIC: cooked DWI; AAIC: cooked AAI; GalA: galacturonic acid; Rha, rhamnose; Gal: galactose; Man: mannose; Ara: arabinose; Glc: glucose; Fuc: fucose; Xyl: xylose. “---” indicates that the monosaccharide is not detected or the sugar ratio of the component is not calculated.

Treatment		Sugar Content (mg/g)								Sugar Ratios		
		Man	Rha	GalA	Glc	Gal	Xyl	Ara	Fuc	1	2	3
WSF	DWI	0.88 ± 0.02 b	8.81 ± 0.31 c	75.05 ± 0.42 a	3.22 ± 0.04 b	17.41 ± 0.22 c	2.99 ± 0.04 b	17.30 ± 0.13 c	0.84 ± 0.06 a	1.583 ± 0.015 a	0.117 ± 0.004 d	3.941 ± 0.114 b
	AAI	0.50 ± 0.14 c	11.90 ± 0.07 b	72.07 ± 0.09 b	3.52 ± 0.07 a	22.97 ± 0.16 b	4.79 ± 0.07 a	22.50 ± 0.26 b	0.80 ± 0.05 a	1.143 ± 0.006 b	0.165 ± 0.001 c	3.819 ± 0.018 b
	DWIC	1.02 ± 0.03 b	6.42 ± 0.05 d	21.91 ± 0.23 c	1.47 ± 0.07 c	13.15 ± 0.12 d	0.74 ± 0.02 c	11.72 ± 0.10 d	0.76 ± 0.08 a	0.667 ± 0.006 c	0.293 ± 0.005 b	3.876 ± 0.057 b
	AAIC	4.61 ± 0.29 a	15.28 ± 0.55 a	17.74 ± 0.01 d	3.12 ± 0.07 b	32.65 ± 0.06 a	—	37.32 ± 0.06 a	0.35 ± 0.01 b	0.210 ± 0.000 d	0.863 ± 0.032 a	4.580 ± 0.156 a
CSF	DWI	0.08 ± 0.01 c	8.37 ± 0.03 a	45.12 ± 0.40 a	2.40 ± 0.02 b	16.99 ± 0.33 a	2.90 ± 0.21 a	18.06 ± 0.05 a	0.36 ± 0.31	0.967 ± 0.021 c	0.186 ± 0.002 a	4.184 ± 0.033 c
	AAI	0.20 ± 0.01 b	0.11 ± 0.01 c	3.91 ± 0.06 b	3.86 ± 0.10 a	0.11 ± 0.02 c	0.23 ± 0.01 b	0.39 ± 0.03 c	—	4.637 ± 0.140 a	0.029 ± 0.001 c	4.378 ± 0.120 bc
	DWIC	0.21 ± 0.01 a	0.05 ± 0.01 d	0.41 ± 0.11 d	0.51 ± 0.14 d	—	0.24 ± 0.01 b	0.94 ± 0.01 b	—	0.340 ± 0.092 d	0.131 ± 0.030 b	18.058 ± 1.060 a
	AAIC	—	0.16 ± 0.01 b	2.55 ± 0.07 c	1.08 ± 0.14 c	0.47 ± 0.06 b	0.13 ± 0.06 b	0.39 ± 0.08 c	—	2.230 ± 0.202 b	0.060 ± 0.001 d	5.453 ± 0.593 b
NSF	DWI	0.94 ± 0.04 c	6.97 ± 0.02 a	29.44 ± 0.29 a	1.04 ± 0.02 a	18.13 ± 0.09 a	1.81 ± 0.07 a	20.42 ± 0.15 a	—	0.620 ± 0.010 b	0.237 ± 0.002 c	5.536 ± 0.037 b
	AAI	0.73 ± 0.31 c	3.06 ± 0.02 c	15.26 ± 0.14 b	0.63 ± 0.05 b	7.67 ± 0.06 b	0.92 ± 0.05 b	10.25 ± 0.18 b	—	0.697 ± 0.015 a	0.200 ± 0.003 c	5.860 ± 0.078 a
	DWIC	4.19 ± 0.23 a	3.20 ± 0.06 c	1.80 ± 0.22 d	0.98 ± 0.02 a	2.25 ± 0.01 d	—	2.36 ± 0.10 d	1.05 ± 0.02	0.290 ± 0.070 c	1.797 ± 0.193 a	1.440 ± 0.024 d
	AAIC	2.34 ± 0.14 b	3.79 ± 0.27 b	4.40 ± 0.06 c	0.92 ± 0.17 a	3.67 ± 0.54 c	0.47 ± 0.02 c	6.92 ± 0.40 c	—	0.297 ± 0.021 c	0.860 ± 0.070 b	2.807 ± 0.365 c
HF	DWI	1.94 ± 0.18 c	4.75 ± 0.04 c	12.20 ± 0.04 a	129.20 ± 0.63 d	36.47 ± 0.20 b	45.17 ± 0.17 b	35.48 ± 0.22 b	2.95 ± 0.06 b	—	—	—
	AAI	2.64 ± 0.06 b	6.45 ± 0.06 a	9.82 ± 2.29 b	138.52 ± 1.19 c	44.17 ± 0.30 a	72.60 ± 0.62 a	68.63 ± 0.59 a	7.64 ± 0.16 a	—	—	—
	DWIC	1.51 ± 0.06 d	4.87 ± 0.02 b	7.42 ± 0.13 c	224.13 ± 0.39 a	24.57 ± 0.07 c	34.45 ± 0.14 c	24.84 ± 0.02 d	2.09 ± 0.07 c	—	—	—
	AAIC	3.53 ± 0.05 a	3.06 ± 0.02 d	7.03 ± 0.68 c	209.49 ± 0.22 b	16.27 ± 0.03 d	21.37 ± 0.09 d	28.40 ± 0.07 c	1.37 ± 0.06 d	—	—	—

## Data Availability

Data will be made available on request.

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
