# Peer review of "Acetic Acid Immersion Alleviates the Softening of Cooked Sagittaria sagittifolia L. Slices by Affecting Cell Wall Polysaccharides"

_foods, 2023, doi:10.3390/foods12030506_

Round 1

Reviewer 1 Report

The job is quite interesting.The authors used various methods to evaluate the effect of Sagittaria sagittifolia L. immersion in acetic acid on changes in the structure of the cell wall polysaccharides.

They wanted to demonstrate that this would result in a product with better textural properties. Therefore, I believe that determining only the hardness in the texture analysis is insufficient. In the discussion in point 3.1, the authors should refer to the results obtained from other studies and justify the changes that occurred during the processing of Sagittaria sagittifolia L. I would see this point at the end of the manuscript, in a general discussion of the changes that have occurred during the processing of Sagittaria sagittifolia L. Simply stating that something has more or less hardness in a scientific article is not enough.

Pinku 2.2 Please explain in detail what the modification of the method used by the authors in relation to Zhao et al. or an explanation of your methodology in detail. The article is not very long, so please complete this part.

Section 2.10 Where has the analysis of correlation between parameters been used in the discussion? However, it may be worth checking how the texture parameters correlate with other results.

Reviewer 2 Report

The authors present an interesting study concerning the protectio of hardnessn by acid acetic immersion, during the cooking of arrowhead bulbs. The work involves a number of analyses which complement the information and findings. The following comments should improve the manuscript:

*Abstract (Line 12): the expression "Different treatment" can be replaced by a brief description of the treatments in order to show the precise heat treatment and acetic acid concentration.

*Introduction Section:
-Most of people do not know the arrowhead bulb and its use in different dishes in cuisine. To improve the downloads, views and cites of this work, I recommend to extend the description of the plant and respond to the following question: why is it consumed cooked instead of raw?
-Line 45: It seems that Zhao et al. fully described the effect of acetic acid as a pretreatment on cell walls. Please add more details of this research, explaining findings already reported and the study material.
-Line 65: why the study can help improving thermally processed starchy vegetables?? Please explain how rich is arrowhead in this biopolymer?

*Results and discussion:
-Throughout this section, authors mentioned that their findings agreed with previous studies. Readers can be confused whether these "previous studies" come from the same authors of the paper. I suggest to change these phrases by using "other studies" and their variants.
-Section 3.3 (Lines 225 - 228): this sentence corresponds to methods section. Please replace it where it corresponds.
-Figure 5: the X-axis is not homogeneous; please use "retention time" instead of "rotation time" for all graphs. Also, numbers I, II and III must be defined in the caption.

*Few misspellings were found:
-Line 71: use "in" instead of "into".
-Line 141: use "mL" instead of "ml".
-Line 244: use "demonstrated" instead of "emonstrated"
